# Regulation of cGAS Activity and Downstream Signaling

**DOI:** 10.3390/cells11182812

**Published:** 2022-09-08

**Authors:** Bhagwati Joshi, Jagdish Chandra Joshi, Dolly Mehta

**Affiliations:** Department of Pharmacology and Regenerative Medicine, College of Medicine, University of Illinois, 835 S Wolcott Avenue, Chicago, IL 60612, USA

**Keywords:** cGAS, STING, calcium, autoimmunity, inflammation

## Abstract

Cyclic GMP-AMP synthase (cGAS) is a predominant and ubiquitously expressed cytosolic onfirmedDNA sensor that activates innate immune responses by producing a second messenger, cyclic GMP-AMP (cGAMP), and the stimulator of interferon genes (STING). cGAS contains a highly disordered N-terminus, which can sense genomic/chromatin DNA, while the C terminal of cGAS binds dsDNA liberated from various sources, including mitochondria, pathogens, and dead cells. Furthermore, cGAS cellular localization dictates its response to foreign versus self-DNA. Recent evidence has also highlighted the importance of dsDNA-induced post-translational modifications of cGAS in modulating inflammatory responses. This review summarizes and analyzes cGAS activity regulation based on structure, sub-cellular localization, post-translational mechanisms, and Ca^2+^ signaling. We also discussed the role of cGAS activation in different diseases and clinical outcomes.

## 1. Introduction

Pathogenic invaders, including bacteria and viruses, and genetic mutations associated with immunogenic diseases continuously challenge host cell homeostasis [1].A defining feature of innate immunity by the infected cell is to liberate DNA/RNA to induce the cell death pathway through the pattern recognition receptors (PRR) [2,3]. Sun et al. initially identified cyclic GMP–AMP synthase (cGAS) as one of the most crucial cytosolic DNA sensors [4]. Since then, our knowledge of how the liberated DNA/RNA in the host cell provokes an immune response, particularly for the cGAS and its effector, stimulator of interferon genes (STING), has burgeoned. cGAS is now one of the most intensely studied areas of basic and clinical research leading to publications of several excellent reviews [5,6]. As cGAS is one of the most rapidly diverging genes in the mammalian genome [7,8], more understanding of cGAS function and downstream signaling is required. 

To date, our understanding of cGAS structure comes from inactive human enzymes [9,10,11], mouse cGAS (mcGAS), and mammalian homologs [11,12,13,14,15]. We discussed the variability in cGAS sequence within different species and how it dictates cGAS overall function. 

Another rapidly emerging and debatable area in cGAS biology is its sub-cellular localization. The cytosolic cGAS protein pool regulates immune signaling against aberrant dsDNA in the cytosol while the nuclear cGAS pool is inactive [16,17]. These findings suggest that cGAS enzymatic activity is an exclusive function of cytosolic cGAS. We have attempted to decode the functional and physiological aspects of cGAS in the different cellular compartments and the potential upstream cellular stress signals which can influence cGAS sub-cellular localization.

Recent reviews discussed the role of Ca^2+^ signaling in initiating STING-mediated innate immune response [18]. However, how Ca^2+^ signaling intercepts with cGAS activation is still unclear. Thus, we have leaped through published and unpublished data to interlink Ca^2+^ signaling with cGAS activation.

Herein we describe the current understanding of the cGAS activity regulation based on (1) structure, (2) sub-cellular localization, (3) post-translational mechanisms, and (4) Ca^2+^ signaling. We also discuss the role of cGAS activation in different diseases and clinical outcomes. 

## 2. cGAS

Cyclic guanosine monophosphate-adenosine monophosphate (cGAMP) synthase (cGAS), otherwise known as C6ORF150 or Mab-21 domain having 1 (MB21D1), belongs to the nucleotidyltransferases family. cGAS was first identified almost a decade ago by Sun et al. [4]. cGAS activates immune response when it binds dsDNA liberated from various sources such as mitochondrial DNA, micronuclei, self-DNA, viral DNA, bacterial DNA, or the DNA from dying cells [4,11,14,19,20]. In addition to cGAS, several other potential DNA sensors, like DAI, RNA polymerase III, IFI16, DDX41, and DNA helicases, also induce type 1 interferon [21,22,23,24,25,26,27,28]. However, except for cGAS, none of these sensors have been acknowledged universally [29]. 

cGAS binds dsDNA in a length-dependent manner to influence immune responses [13,30]. While the shorter DNA (~20 bp) can bind cGAS, it cannot dimerize and efficiently activate cGAS. The longer dsDNA of >45 bp can dimerize cGAS and form a ladder-like network, leading to maximum enzyme activity [24,31,32,33]. Mechanisms by which dsDNA activates cGAS remain to parse out. However, dsDNA-mediated conformational changes in cGAS activate the enzymes leading to the synthesis of cGAMP [4,14,19,34]. The cGAMP binds STING [34,35,36], which in turn triggers activation of the interferon regulatory factor 3 (IRF3) and nuclear factor-kB (NF-kB) transcription factors [4,20,34]. Activated IRF3 and NFkB induce inflammation through generating type I interferon IFN-β [34]. Thus, dsDNA binding to cGAS and generation of cGAMP induces inflammatory signaling by activating STING. 

### 2.1. Structural Domain of cGAS 

cGAS is ubiquitous but rapidly diverging [7,8]. We analyzed cGAS protein sequences from five common vertebrates (>NP_612450.2 cGAS-HUMAN, >NP_775562.2 cGAS-MOUSE, >NP_001305104.1 cGAS-MONKEY, >sp|I3LM39.1|CGAS-PIG, >XP_038539891.1 cGAS-DOG) using BlastP (Basic Local Alignment Search Tool, NCBI, USA) and found considerable differences in the sequence homology (Figure 1). The h-cGAS shares < 60% amino acid identity with the mouse (m)-cGAS [37] while 70–80% similarity with other vertebrates. Multiple sequence alignment for the corresponding, truncated amino acid sequences illustrated the highly conserved regions throughout the homologs (Figure 1). 

The alignment of the amino acid sequence of cGAS within different mammalian species further suggested that the cGAS N-terminal domain (~160-amino-acids) is highly diverse and disordered. Therefore, the structural and functional characterization of the N-terminal domain of cGAS is highly unpredictable (Figure 1). Recent studies have shown that the serine (13, 37, 64, 129, 143) residue in cGAS-N terminus sense genomic/chromatin DNA but not mitochondrial DNA [38,39]. Unphosphorylated cGAS-N-terminus seem to form multivalent interactions with the nuclear chromatin DNA. However, hyperphosphorylation of serine and threonine residues (Ser13, Ser37, Ser64, Thr69, Thr91, Ser116, Ser129, and Ser143) within N-terminus blocked chromatin DNA sensing in mitotic cells [39]. Evidence indicates that the N-terminus of cGAS in humans has a phosphoinositide-binding domain and interacts selectively with phosphatidyl inositol (4,5) bisphosphate (PIP2) in the plasma membrane, and this interaction restrains cGAS activity from self-DNA [40]. Consistently, cGAS deprived of N-terminus (δN-cGAS) localizes to the cytosolic and nuclear compartments [40]. Thus, the non-enzymatic N-terminal domain can dictate cGAS localization in the cytosol, or on centromeres leading to activation of nuclear cGAS [41]. However, Zhou et al., showed that two unique substitutions K187 and L195 in the N-terminus of hcGAS are responsible for human-specific control of 2′3′ cGAMP synthesis [42].

The C terminal catalytic domain of cGAS consists of two structurally strong-conserved motifs, which include the nucleotidyltransferase (NTase) core domain (160–330) and a Mab21 domain (213–513) [11,13]. The NTase domain harbors the conserved acidic residues, which function as nucleic acid sensors but are indispensable for cGAS enzyme activity [4,43]. The Mab 21 domain of cGAS is the catalytic domain and harbors the conserved Zn finger motif (H390-C405), which has a weaker binding affinity for DNA than NTase but is vital for cGAS enzymatic activity and downstream innate-immune signaling activation [11,13]. Substitutions to the zinc coordination motif (H390A, C396A, C397A, C404A) abolish cGAS activity when exposed to ds DNA [38]. 

Free cGAS (not bound to dsDNA) does not have a suitably structured catalytic site [9,13,14]. The binding of dsDNA (length > 45 b) between two cGAS monomers creates a structured catalytic site [11,12,44], leading to cGAS activation [13,14]. While amino acids required for cGAS activity are conserved within species, h-cGAS exhibit severely reduced levels of cGAMP synthesis because of enhanced selectivity to DNA, with activity increasing with dsDNA of >45 base pairs [24,32]. h-cGAS, therefore, balances cGAS sensitivity to pathogen-or stress-derived DNA while allowing accurate tolerance of self-DNA. Thus, N and C terminus control cGAS activity upon binding dsDNA. 

### 2.2. cGAS Localization

The subcellular localization of cGAS is still a matter of debate because cGAS localizes in the plasma membrane, cytosol and nucleus [40,41,45,46]. Cytosolic cGAS may be beneficial for triggering genotoxic stimulation with the DNA that needs restraining in the nucleus [40,41]. Plasma membrane-bound cGAS is translocated to the cytosol upon stimulation with exogenous DNA [40]. The association of cGAS with the plasma membrane is mainly due to electrostatic interactions between the positively charged N-terminal domain of cGAS (residues 1 to 160) and the negatively charged PIP_2_ [40]. The unstructured positively charged N-terminal domain of cGAS can bind the extracellular dsDNA, but there is a lack of enough evidence of activation of cGAS after binding of DNA to the N-terminal domain [39] (Figure 2). However, cGAS lacking N-terminal domain strongly induces interferon responses to genotoxic stress but weaker responses to any bacterial or viral infection [40] (Figure 2). Gentili et al., 2019, have shown that nuclear cGAS is required for constitutive expression of interferon stimulating genes (ISGs), suggesting that a basal level of nucleic acids activates the sensor in the absence of microbial infection or apparent damage [41,47] (Figure 2). Thus, the constitutive induction of beneficial (early) type I IFN response confers antiviral protection to the host cells [48,49,50,51]. Also, cGAS can be activated differentially during mitotic errors and later contributes to birth defects, aging, carcinogenesis, and cancer therapy [52,53]. While mitotic error occurs at a low frequency in normal cells, a much higher frequency occurs in conditions, such as chemotherapeutics which can lead to cGAS activation via forming micronuclei and chromatin bridges [54,55]. Thus, the localization of cGAS at different compartment of the cells mediate specific activity and the ability to respond distinctly between foreign and self-DNA. 

### 2.3. cGAS Droplet Formation

DNA arising in the cytoplasm binds cGAS and drives the formation of cytoplasmic liquid phase condensation called “droplets or foci” [56]. The DNA and RNA can also form liquid droplets. However, only DNA binding will induce a conformational change to activate cGAS [57]. While the molecular mechanism and functional effect of such cGAS-DNA droplet formation are poorly understood, these foci lack a membrane and can fuse to form larger foci, a feature of liquid drop without a membrane [58]. These droplets function as microreactors in which the enzyme (cGAS) and reactants (DNA) are concentrated in the presence of ATP and GTP to enhance the production of a product (cGAMP) (Figure 2). This mechanism allows cGAS to detect the presence of DNA in the cytoplasm above a certain threshold due to the multivalent interaction of cGAS positive charge with negatively charged DNA. However, cGAS-DNA interaction is vulnerable to cytoplasmic salt concentrations, which may prevent spurious activation of cGAS by self-DNA below a certain threshold. Du and Chen [56] showed that Zn^+^ ions substantially enhance cGAS phase separation with DNA and its enzymatic activity at physiological salt concentrations. Free zinc ions are stored mainly in organelles such as the mitochondria and the endoplasmic reticulum [59], indicating that Zn^+^ delivery to the cytosol may be critical for regulating cGAS activity in cells. Further characterization of the dynamics and composition of the cGAS condensates should provide deeper insights into the mechanism by which Zn^+^ levels in mitochondria or ER contribute to regulating cGAS activity while simultaneously avoiding autoimmune reactions to self-tissue. Altogether, cGAS droplet formation explains the cytosolic regulation of cGAS after sensing DNA and reveals a mechanism that enables the cellular balance between tolerance and innate immune activation. 

## 3. Regulation of cGAS Activity

cGAS serves as a node for regulating diverse regulatory functions in response to various cellular processes including autophagy, apoptosis, inflammation, tissue fibrosis, pathogenic infections (bacteria or virus), tumorigenesis, and cancer [19,53,60,61,62,63]. Dysregulation of cGAS results in intrusion of cellular homeostasis during viral or bacterial infections, creating a tumor-prone microenvironment by promoting evasion of immune surveillance. On the other hand, hyperactivated cGAS signals autoimmune disease but may serve as subsidiary cancer therapy. Thus, the tight regulation of cGAS signaling is highly required to maintain cellular homeostasis, preventing thereby autoimmune and pathological angiogenesis. 

### 3.1. Transcriptional Regulation of cGAS 

The transcriptional regulation of cGAS activation is not a well-explored area. cGAS promoter contains three IFN-sensitive response elements (ISREs) and one STAT1 binding site that can potentially induce cGAS synthesis by type 1-IFN [57]. As h-cGAS and m-cGAS are <60% homologous, cGAS can be synthesized differentially in different species. Chen et al., characterized the molecular mechanisms controlling the cGAS transcriptional activity [64]. By a series of 5′ deletion and promoter constructions, they found that the region (−414 to +76 relatives to the transcription start site) was sufficient for promoter activity [65]. Mutation of SP1 and CREB binding sites in this promoter region led to an apparent reduction of the cGAS promoter activity [65]. Moreover, elevated expression of microRNA (miR)-93/25 observed in breast cancer facilitates tumor evasion from immune surveillance and destruction partially through downregulating cGAS expression. Thus, miR, SP1 and CREB may control cGAS synthesis and inflammatory signaling.

### 3.2. Translational Regulations of cGAS Activation

Posttranslational modifications such as phosphorylation, ubiquitination, sumoylation, and acetylation play a pivotal role in cGAS signaling regulation, dimerization, and transcriptional regulation, ultimately regulating a plethora of biological processes such as cell survival and DNA damage (DDR) in response to genomic insult (Figure 3).

#### 3.2.1. Phosphorylation

cGAS contains many serine/threonine residues within its catalytic sites and C-terminus [66]. Phosphorylation of cGAS by serine kinases such as serine/threonine-protein kinase (AKT), cyclin-dependent kinase-1 (CDK1), and Aurora kinase B (AKB) has been shown to repress cGAS activity and thereby immune response [39,67,68]. Seo et al. showed that HSV-1 infection on 293T cells leads to AKT activation which in turn phosphorylated cGAS at S291 (m-cGAS) and S305 (h-cGAS) [67]. cGAS phosphorylation at S291/S301 impaired cGAS generation of cGAMP and thereby the production of type 1 IFN [67]. In another study, HSV-1 infection of human primary fibroblasts and HEK293T cells caused cGAS phosphorylation at multiple serine sites (Ser37, Ser116, Ser201, Ser221, Ser263) which also suppressed cGAS activity during their post-translational modification, however, the signaling behind the phosphorylation is not yet clearly understood [66]. Zhong et al., showed similar suppression of cGAS activity after phosphorylation at S305 residue by CDK1-cyclin B kinase complex in mitotic cells [68]. Li et al., recently demonstrated that hyperphosphorylation of cGAS N-terminus at serine and threonine residues (Ser13, Ser37, Ser64, Thr69, Thr91, Ser116, Ser129, and Ser143) by AKB blocked chromatin DNA sensing in mitotic cells [39]. Thus, it is plausible that AKT, CDK1, and AKB govern cGAS phosphorylation to control cGAS activation during critical phases of the cell cycle like DNA replication (S) and mitotic division (M). However, cGAS phosphorylation by these kinases was reversed either by protein phosphatase 1 (PP1) [68] or PP6 [69] reinstating cGAS DNA-sensing ability. 

Interestingly, cGAS also contains several tyrosine residues of unknown functions. A study showed that the phosphorylation of Tyr215 in hcGAS by BLK (B lymphocyte kinase) retained cGAS in the cytoplasm for cytosolic DNA sensing [60]. Intriguingly, we showed that synthetic dsDNA as well DNA isolated from Pseudomonas aeruginous and HSV-1 infection of vascular cells induced cGAS phosphorylation at Y215 and Y242 residues by the tyrosine kinase, cSRC (data not published). We further showed that cGAS phosphorylation at Y215/Y242 promoted cGAS-DNA liquid phase condensation and cGAS activation since transduction of mutated cGAS in endothelial cells inhibited these events (unpublished). Thus, serine and tyrosine phosphorylation of cGAS seems to have an antagonistic role in regulating cGAS activity.

#### 3.2.2. Ubiquitination

Several ubiquitin ligases are shown to regulate cGAS activity. The E3 ligase, TRIM56, monoubiquitinated cGAS at Lys335 which resulted in a marked increase in its dimerization, DNA-binding activity, and cGAMP production [70]. TRIM41 also promoted cGAS monoubiquitination using biochemical assays [71]. Wang et al., reported that ER ubiquitin ligase, RNF185, specifically catalyzed the K27-linked poly-ubiquitination of cGAS at K-384 and promoted its enzymatic activity [72]. The patients suffering from Systemic Lupus Erythematosus (SLE), has constitutive activation of cGAMP-STING signaling [73], and displayed elevated expression of RNF185 mRNA [72]. Whether increased RNF185 mRNA was the factor in increasing cGAS-STING signaling is unclear. It was reported that upon DNA virus infection, USP14 was recruited by TRIM14 to remove K48-linked ubiquitin chains at the Lys414 site in h-cGAS, leading to cGAS stabilization to promote antiviral innate immunity [34]. Similarly, USP27X [74,75] and USP29 [51] have also been reported to stabilize cGAS proteins by cleaving K48-linked poly-ubiquitin chains from cGAS, and both serve as positive regulators in activating innate immunity to fight against DNA viral infection. Thus, deubiquitinase and ubiquitinase have additional control on cGAS activation. 

#### 3.2.3. SUMOylation

Like ubiquitination, TRIM38 maintains SUMOylation of m-cGAS at Lys217 and Lys464 residues (corresponding to Lys231 and Lys479 in hcGAS), which prevented K48-linked cGAS polyubiquitination and degradation [75]. On the other hand, SENP7 alleviates SUMO-mediated suppression of cytosolic DNA sensing by removing cGAS SUMOylation [76].

#### 3.2.4. Acetylation

The acetylation of cGAS can regulate cGAS activity either positively or negatively depending upon the acetylation sites. The protein acetylation occurs on lysine residues by the action of acetyl-transferases and deacetylases [77]. Dai et al., found acetylation of hcGAS at Lys384, Lys394, and Lys414 residues at resting state which restrained cGAS activity [78]. However, deacetylation of these sites upon DNA challenge enabled cGAS activation [78]. Thus, this group introduced a promising role of aspirin, which could directly acetylate cGAS to inhibit cGAS activation, as a therapeutic strategy in treating autoimmune diseases, such as Aicardi–Goutieres syndrome (AGS) [66,78,79]. Recently, the N-terminal domain of human cGAS was reported to be acetylated by lysine acetyltransferase, KAT5, at Lys47/Lys56/Lys62/Lys83 residues. These acetylation events increased cGAS binding with DNA and activity in response to DNA challenge [79]. Further functional validation suggests that acetylation at Lys414 suppresses, while acetylation at Lys198 promotes h-cGAS activation. Interestingly, h-cGAS-Lys198 acetylation was found to be decreased by quantitative proteomics upon infection by either HSV-1 or HCMV (human cytomegalovirus), suggesting that these DNA viruses might evade innate-immune surveillance by hijacking acetylation of cGAS at these sites [66]. 

Thus, phosphorylation, ubiquitination, SUMOylation, and acetylation all control cGAS activity. How these posttranslational modifications of cGAS are regulated at different stages of infection and whether ubiquitination precede acetylation or vice versa may provide an approach to specifically target cGAS activity while preserving host-defense. 

## 4. cGAS Activation of STING 

cGAS-induced cGAMP binds STING (stimulator of interferon genes) in the ER (endoplasmic reticulum) membrane to induce inflammatory signaling [46,67] (Figure 4). STING is a transmembrane homodimer that upon being activated phosphorylates itself and translocates to the Golgi apparatus where it binds to and activates TANK-binding kinase 1 (TBK1) and IFN regulatory factor 3 (IRF3) via a phosphorylation-dependent mechanism, leading to the generation of type 1 IFN [34,80]. 

### 4.1. Role of Gasdermin D

Gasdermin D (GSDMD) forms ~21 nm diameter oligomeric pores in the plasma membrane which induces cell death (pyroptosis) [81]. Studies showed that caspases (caspase-1 and -4/-5/-11) cleaved human and mouse GSDMD after Asp275 and Asp276 respectively, generating an N-terminal cleavage product (GSDMD-N) [82]. GSMD-N triggered pyroptosis and the release of inflammatory cytokines IL-1β and IL-18 and downstream signaling [83]. Interestingly, Huang et al., showed that GSDMD-N could form pores in the mitochondria to release mitochondrial DNA (mtDNA) that activates the cGAS-STING pathway to suppress cellular proliferation [84]. A recent study also showed that PIP2 content in the membrane retains GSDMD pores predominantly in the open state [85]. Since PIP2 induces cGAS translocation [40] it is possible that membrane PIP2 levels control both GSDMD and cGAS activation and thereby STING signaling (Figure 4). Thus, GSDMD and cGAS may operate in parallel to trigger STING signaling [86].

### 4.2. Role of Ca^2+^ Signaling

ER also forms the hub for regulating a rise in intracellular Ca^2+^ signaling. An increase in cytosolic Ca^2+^ is required for several primary cellular processes including apoptosis, proliferation, cellular motility, differentiation and exocytosis, and innate immunity [87,88,89]. Interestingly, Ca^2+^ signaling seems to have a dual role in regulating cGAS-STING signaling since STIM1 (stromal interaction molecule 1), an established sensor of Ca^2+^ in ER physically interacts with STING via its EF-hand domain and inhibits STING activity by both cGAMP and cGAMP independent STING activation [90]. Similarly, in systemic lupus erythematosus (SLE) hyperactivation of cGAS-STING signaling in T cells also required suppression of Ca^2+^ homeostasis, but whether STIM1 was involved is not clear [91]. In contrast, the DNA-dependent activator of IFN-regulatory factors (DAI), which functions as a secondary cytoplasmic DNA sensor to cGAS, induces NF-κB and IRF3 in a calcium-dependent manner [18,92]. Likewise, the quenching of cytosolic Ca^2+^ by BAPTA-AM (a Ca^2+^ chelator) or CGP37157 (which inhibits the mitochondrial sodium-Ca^2+^ pump) dampened DAI-induced autoimmune response in macrophages [86]. 

STIM1 is a single-pass transmembrane protein and binds Ca^2+^ in the lumen of ER in several cell types such as T cells and endothelial cells. STIM1 organizes as puncta following ER-depletion and binds to plasmalemmal Ca^2+^ channels such as ORAI1 and TRPC1/TRPC4 at ER-PM junctions to mediate the influx of Ca^2+^ from the extracellular space into the cell by a process referred to as store-operated Ca^2+^ entry (SOCE) [93,94,95]. STIM1 inhibits STING in macrophages by anchoring it to the ER, which inhibits its translocation to the Golgi and thus halts STING-mediated IFN-β production. However, we showed that in alveolar macrophages, the most sentinel cells in the tissue, STIM1 is dispensable since endotoxin stimulated Ca^2+^ influx in a STIM independent but in transient receptor vanilloid potential (TRPV4) channel-dependent manner [96]. 

Ca^2+^ is known to activate calcium/calmodulin-dependent protein kinase II (CAMKII), which may alter cGAS activation of STING. CAMKII can bypass cGAS activity by activating STING via inducing the activity of AMP-activated protein kinase (AMPK) [97]. AMPK has been shown to induce STING signaling through repressing ULK1 [98,99]. ULK1 phosphorylates and negatively regulates cGAS-STING signaling [99] (Figure 4). However, ionomycin-induced saturating levels of cytosolic Ca^2+^ inhibited STING activity indicating that an optimal Ca^2+^ influx may be required for STING function [100]. PIP2 can suppress cGAS activity through AKT phosphorylation of cGAS [67]. Thus, there is a need to tease out whether Ca^2+^ influx promotes or dampens cGAS activity. 

In mouse embryonic fibroblasts and human Jurkat T cells, loss of STING leads to elevated Ca^2+^ influx [101]. While in the absence of STING, STIM1 became enriched at ER-PM sites and this likely led to the dysregulated induction of store-operated Ca^2+^ entry. Thus, it is possible that the STIM1-STING binding under resting conditions reciprocally regulates the pool of STIM1 protein to maintain Ca^2+^ homeostasis. These findings also suggest that during STING activation, STIM1 may be free to translocate and interact with cGAS or ORAI1, resulting in a transient cytosolic Ca^2+^ influx. Future work studying this relationship will be important in enhancing our understanding of the interconnecting pathways linking innate immune signaling and Ca^2+^ homeostasis. However, the activation of cGAS-STING signaling during calcium influx and calcium-related signaling is yet to be explored.

## 5. cGAS-STING Pathway and Pathologies 

cGAS-STING signaling is expressed in various immune cells, including macrophages, dendritic cells, T cells, epithelial cells, endothelial cells, and even some specific fibroblasts (Barber, 2015). cGAS-STING signaling is emerging as a significant regulator of pathogenic infection, development of cancer or autoimmune diseases, or inflammatory disorders. Below we summarize cGAS-STING role in regulating these diseases. 

### 5.1. Acute Lung Injury (ALI) and COVID-19

ALI is a devastating inflammatory condition of the lungs that is characterized by disruption of alveolar-capillary junctions leading to an uncontrolled influx of proteins rich fluid and blood cells into the alveolar space [102,103,104,105]. Neutrophils and tissue-resident macrophages propagate inflammatory responses by releasing various inflammatory cytokines and chemoattractants. However, these cells also seem to have a resolving role in ALI [106]. Upon sensing pathogens macrophages orchestrate the inflammatory signaling by releasing inflammatory cytokines leading to the influx of activated neutrophils [107]. Later these macrophages clean up the debris by a process called efferocytosis to resolve the lung inflammation [108,109]. Thus, based on their activation lung macrophages can acquire inflammatory (M1) or anti-inflammatory (M2) lineage [110,111]. The alveolar macrophages (AM), the major immune cells in the lungs, are the first offender of invading pathogens. Another subset of macrophages in the lungs are interstitial (IM) and monocytes-derived macrophages (MoMacs). While the role of IM remains unclear, MoMacs compensate for the dying AM and serve as an anti-inflammatory and pro-resolving macrophage. 

IFN-β is one of the important cytokines in macrophages induced by viruses as well as bacteria [112,113]. IFN-β can be generated either by non-canonical (inducible signaling pathway) or canonical (signaling which is constitutively active) pathway and seem to dictate the outcome of ALI and patient mortality from COVID-19 as well as autoimmune diseases such as Aicardi-Goutieres syndrome (AGS), Systemic lupus erythematosus (SLE), and STING-associated vasculopathy with onset in infancy (SAVI), etc [114,115,116,117,118,119]. The initial induction of type I IFNs limits virus propagation, however, a sustained increase in IFN-β level in the later stage of the infection is associated with aberrant inflammation. Studies showed that type I &type III IFNs significantly precipitated the SARS-CoV-2 replication and pathological responses associated with SARS-CoV-2 and a crucial component of a skin lesion in these patients [120,121,122,123].

We showed that conditional ablation of MoMacs during injury augmented the activation of cGAS-STING signaling by LPS and *Pseudomonas aeruginous* (PA) [103]. In WT mice LPS and PA resulted in reversible generation IFNβ leading to resolution of ALI. However, ablation of MoMacs during injury increased the production of secondary messenger, cGAMP in the lungs leading to persistent STING signaling and lung injury [103]. While the detailed mechanism remains to be parsed out, these authors showed that MoMacs inhibit cGAS-STING signaling by generating a lipid mediator, sphingosine 1 phosphate. Consistently, the adoptive transfer of MoMacs in mice lacking endogenous MoMacs reduced IFN-β generation and resolved ALI. 

In another study, Hong et al. showed that accumulated mtDNA can stimulate cGAS in endothelial cells leading to the expression of type I IFNs and cell death [84]. They showed that LPS activated gasdermin D, which formed mitochondrial pores to enhance the influx of mtDNA into the cytosol of endothelial cells [84]. mtDNA was recognized by the DNA sensor cGAS leading to the generation of the second messenger cGAMP, which suppressed endothelial cell proliferation by downregulating the signaling induced by the transcription factor, YAP1. Studies have also shown that in idiopathic pulmonary fibrosis patients (mt DNA) can be accumulated in type II alveolar epithelial cells due to ER stress and PINK 1 deficiency [124,125,126,127]. In COPD patients increased level of cell-free DNA in the buccal cell was associated with apoptotic or necrotic cell death [127,128]. It is possible therefore that mtDNA accumulation in Type II cells can exacerbate lung damage as these cells generate Type I epithelial cells to repair lung injury.

Using a lung-on-chip model, Domizio et al., showed that upon infection with SARS-CoV-2, macrophages released mitochondrial DNA which activated cGAS–STING signaling in endothelial cells leading to their death and irreversible IFN-β production [121]. Pharmacological inhibition of STING reduced severe lung inflammation induced by SARS-CoV-2 and improved disease outcome. In terms of preventive measures during viral infection, early induction of IFN-β before the high viral load was found to be protective, whereas late induction resulted in increased inflammation and lethal pneumonia [15,129]. 

Thus, these findings raise the possibility that cGAS-STING signaling in lung cells can exacerbate ALI, and therefore cell therapy (CD11b+-Mɸ) or STING inhibitors are valuable targets for the treatment of ALI pathology in inflammatory conditions

### 5.2. Adaptive Immunity

Immune system checks on the invading microbes by activating innate or adoptive immunity. As the name suggests innate immunity is antigen and memory-independent, the non-specific first line of defense dependent on pattern recognition receptors (PRRs) (discussed above). In contrast, adaptive immunity is antigen-dependent, specific for the pathogenic antigens, and has immunological memory to kill the pathogens if attacked by the same pathogens. Macrophages and T lymphocytes are the most representative cells of innate immunity and adaptive immunity, respectively. Macrophages, as major professional antigen-presenting cells (APCs), are best known for their ability to prime the host defense by engulfing foreign pathogens [130]. The DNA released through pathogens or endogenous necrotic/apoptotic cells activates the cGAS-STING signal pathway in macrophages, leading to the release of cytokines including IFN-β, which leads to the ani-viral response, to cope with various stresses and maintain homeostasis [131]. Over activation of the cGAS-STING signaling in macrophages can thereby lead to auto-immune diseases [80,132]. For instance, STING-associated vasculopathy in infancy (SAVI), Systemic Lupus Erythematosus (SLE), and Aicardi-Goutières syndrome (AGS) are auto-immune diseases caused by the overproduction of type-I interferon. In SAVI, mutation of STING (R281Q) in patients results in the gain of function of STING causing long-lasting IFN-β production and severe autoinflammatory diseases [133]. AGS is characterized by inherited encephalopathy and leads to severe mental and physical problems, that affect newborn infants. AGS occurs due to a genetic mutation in DNase II, Trex1 (also known as DNase III), or RNaseH2 [134,135,136]. However, in AGS, the mutation in DNAses leads to the amassing of cytosolic DNA resulting in the overactivation of cGAS-STING signaling to generate IFN-β [136,137]. Similarly, in SLE increased IFN-β generation affects various cells and organs such as blood, lung, interstitial, kidney, and skin [138,139,140]. Thus, the cGAS-STING signaling pathway in macrophages and T cell may serve as a connecting link between innate immunity and adaptive immunity [141].

### 5.3. Cancer

cGAS-STING signaling plays a crucial role in regulating various cancer subtypes. Suppression of STING activation in pancreatic cancer cells enhanced tumor size by reducing the infiltration of immune cells [142,143]. In mouse model of the breast cancer, cGAMP reduced tumor burden [144] and liposomal delivery of cGAMP also activated STING in the macrophages [145]. In syngeneic ovarian cancer (ID8) and colon cancer (CT26), poly (ADPribose) polymerase inhibitors (PARPi) increased IRF3 and STING activity [146]. Grabosch et al. found that cisplatin reduced ovarian cancer growth by increasing type I IFN production and inflammation [147]. Xie et al., 2019 found that tumor-associated mutations, G303E and K432T in h-cGAS [148] induced uterine endometrioid carcinoma by suppressing cGAMP generation [149].

Mechanistically, in the tumor microenvironment, STING is widely expressed in multiple cell types such as dendritic cells, myeloid cells, and endothelial cells [150,151,152] and serves as an important pathway for tumor-associated antigen presentation and regression of the tumor in mouse tumor models [153]. Unlike normal cells, cancer cells possess unstable genomes which undergo chromosome mis segregation during mitosis. The consequence of such segregation is the defective generation of micronuclei and their rupture that releases their genomic contents into the cytosol, resulting in activation of the cGAS-STING pathway [132,154,155,156]. Thus, mtDNA from ruptured micronuclei but not cytosolic DNA stimulates the cGAS-STING pathway in cancer. Oxidative stress in cancer cells is the primary mechanism to increase the porosity of the mitochondrial membrane leading to the export of mtDNA in the cytosol [84,157,158]. Indeed, the tumor cells are deprived of mtDNA and import it through exosomal transfer from host cells to restore mitochondrial function which enhances metastatic potential [23,159]. Thus, activation of cGAS-STING, through mtDNA and/or dsDNA, in cancer cells may serve as a checkpoint to the initial progression of neoplasm by the secretion of Type 1 IFNs inflammatory genes. Alternatively, it leads to the release of various other pro-inflammatory cytokines, chemokines, proteases, and growth factors. Collectively these various factors are called senescence-associated secretory phenotype (SASP) and have played a major role in the restriction of tumorigenesis [160,161,162]. Therefore, the STING activation in cancer cells suppresses its growth and/or recruits immune cells to clear cancer cells [160,163,164]. 

Antigen-presenting cells (APCs), such as dendritic cells (DCs) and macrophages, are thought to clear necrotic malignant cells [165,166]. DNA from tumor cells is also inserted into the immune cells such as macrophages and dendritic cells, however, the mechanism of DNA insertion inside the immune cells is not known [153]. Some studies suggest that ATP-gated P2X7 receptors are involved in the delivery of DNA to tumor- associate-macrophages (TAMs) to activate STING signaling in immune cells and enhance the tumor-antigen presentation into the CD8 T cells for their proliferation and cytotoxicity activity [153,167,168]. Activation of STING-signaling in TAM is associated with the polarization of TAM into the M1 stage and suppressing its tumor progressing activity. cGAMP generated into the tumor cells is transported into the host immune cells by SLC19A1, a folate transporter [169], which in turn activates STING-IRF3 signaling in immune cells to induce its tumoricidal activity such as in NK-cells [149].

Studies by various group showed that direct injection of STING agonist into the tumor microenvironment also decrease the suppressive Foxp3+ Treg protein in association with CD8+ T cell activation. DCs in the tumor microenvironment mostly respond to the tumor cells generated cGAMP to activate STING signaling. However, other cells may also respond such as fibroblast cells to inhibit tumor growth [167].

Taken together, cGAS-STING activation in the tumor microenvironment is important for anti-tumor activity and its regression. However, uncontrolled activation of the cGAS-STING pathway can promote angiogenesis due to excessive release of type I IFNs, chemokines, and growth factors [170].

### 5.4. Role of cGAS in Other Diseases

cGAS activation also causes other diseases, including neurological and macular degeneration and liver and renal diseases. Several excellent reviews discussed these topics [171,172,173,174,175,176,177,178]. Therefore, we discussed these diseases briefly in the following section 

In the brain, microglia are the principal cells that generate cGAS-STING dependent type-I interferon [172].

However, excessive cGAS activation leads to microglial pyroptosis after viral infection, trauma, stroke, and subarachnoid hemorrhage causing chronic neurodegenerative diseases [172,177]. Moreover, aging is a crucial factor in precipitating neurological disorders via augmenting cGAS-STING signaling and generation of aged diploid fibroblast cells [172,179,180,181]. In Alzheimer’s patients, oxidative stress induced by the accumulation of tau and beta-amyloid protein aggregates in the neurons leads to cell death and activation of cGAS-STING signaling [178,180,182,183].

In the liver, malfunctioning of kupffer cells or monocytes-derived macrophages and hepatocytes is linked with mtDNA release and cGAS activity causes various liver diseases including hepatic ischemia-reperfusion injury, and hepatic cancer [171,173,184,185]. In this context, Luther’s lab showed that connexin-32 expression in hepatic parenchyma is a significant regulator of cGAS-induced type-I interferon synthesis in alcoholic liver diseases [186]. Iron accumulation in hepatic cells and metabolic stress also activated cGAS-STING signaling resulting in an inflammatory liver [187]. 

Persistent cGAS-STING activation is also a leading cause of kidney injuries such as glomerulonephritis, vasculitis, and chronic kidney injury that can culminate in kidney fibrosis [176,188]. Cell death occurring from post-drug-induced nephrotoxicity, diabetic nephropathy, and ischemia-induced renal injury also activated cGAS-STING activation in the renal tubules [178]. Thus, initial insults in various tissues such as microglia, hepatocytes, and renal tubules can cause neurological, liver, and kidney diseases which can be aggravated further by aging. 

## 6. Clinical Aspect of cGAS

cGAS-STING signaling is an intense area of potential therapeutics. The available cGAS inhibitors suppress signaling either by binding to the cGAS active site or competing with dsDNA. ODN-A151, a small molecule inhibitor of cGAS, competes with dsDNA [189]. Aspirin, an existing non-steroidal anti-inflammatory drug, targets cGAS by acetylating the enzyme at Lys384, Lys394, and Lys494 [5,78]. Other anti-malaria drugs, such as quinine, chloroquine, and hydroxychloroquine, inhibit cGAS activity by binding cGAS at R342 and K372 of the DNA binding site [5,190]. 

Sorafenib, an FDA-approved antitumor drug for hepatocellular carcinoma, a small molecule inhibitor of multiple kinases, inhibits the cGAS-STING signaling induced by RLR-MAVS pathways [191]. Other inhibitors of cGAS include tetrazolo[1,5-a] pyrimidine-7-ol, pyrazolopyrimidinones, and tricyclic benzo-fluoropyrimidine compounds [192,193]. Suramin is being touted in a clinical trial for inhibiting the cGAS activation in human autoimmune disease [194]. RU.521 inhibited macrophage-cGAS signaling in a mouse model of AGS [195,196], but this compound failed in clinical studies due to its very low efficacy against h-cGAS [196]. Chu et al., identified perillaldehyde (PAH), a natural monoterpenoid derived from Perilla frutescens, as an effective inhibitor of cGAS signaling in the preclinical AGS model [197]. Whether PAH is effective in inhibiting cGAS signaling in AGS patient remain unclear.

## 7. Conclusions

We discussed the cGAS-STING signaling as a fundamental process regulating cell response to dsDNA. We also assessed the requirements of an unstructured N terminal and highly organized C-terminal of cGAS in regulating its catalytic activity. Further, we learned that the subcellular localization of cGAS specifically dictates the ability to respond distinctly between foreign and self-DNA. We also described the role of cGAS-DNA induced liquid phase condensation to explain the cytosolic regulation of cGAS after sensing DNA, which reveals a mechanism that enables the cellular balance between tolerance and innate immune activation. We also discussed the tight regulation of cGAS signaling to maintain cellular homeostasis and the prevention of autoimmune and pathological angiogenesis. The post-translational regulation of cGAS during any pathogenic insult may either activate or deregulate the cGAS-STING signaling post-infection. cGAS regulation by PIP2 at the membrane is crucial for cGAS sensing of self-DNA. PIP2 is a major regulator of Ca^2+^ signaling through the ion channels, and the endoplasmic reticulum (ER) stores linked Ca^2+^ -sensor, STIM1. STIM1-mediated Ca^2+^ entry is known to be involved in regulating both innate and adaptive immunity. However, studies indicate that STIM1 can regulate STING signaling independent of Ca^2+^. We also discussed the role of cGAS in various inflammatory disease conditions. The COVID-19 outbreak caused by the newly described SARS-CoV-2-RNA virus caused devastating effects worldwide. Interestingly, the signaling output of SARS-CoV-2 is the transcriptional up-regulation of type-I IFNs and other IRF3 target genes, such as ISGs. 

## 8. Future Directions 

cGAS activity regulates various inflammatory diseases and cancer. Because cGAS cannot discriminate between self and non-self-DNA, a question emerges whether cGAS sensing of self-DNA can be used as a biomarker to identify the initial stages of inflammatory and autoimmune disease. COVID-19 infection causes massive death of lung cells such as macrophages and epithelial cells that may induce self-DNA accumulation. Whether assessing DNA concentration in the bronchoalveolar lavage can be used as a measure of viral load and cGAS activity needs to be seen. Thus, advancing research in these directions may lead to early diagnosis and treatment of life-threatening vascular and lung inflammatory diseases.

We described the role of serine versus tyrosine kinases in regulating cGAS activity. These studies raised a fundamental question of how post-translationally modified cGAS contribute to pathological processes. Aurora kinase B and cSrc are involved in various cancers and other human diseases. A question is whether these enzymes contribute to pathological processes through regulating cGAS modifications. Another interesting question is whether protein tyrosine phosphatases such as PP1 and PP6 function as an inducer of tyrosine phosphorylation of cGAS to promote downstream signaling. In this context, studies need to address whether protein tyrosine phosphatase such as PTP and serine-threonine kinases such as PP1 and PP6 plays a dominant role in regulating the outcome of cGAS downstream signaling. The acetylation and ubiquitination regulate cGAS activity, but whether ubiquitination precedes acetylation remains elusive. 

Membrane PIP2 regulates cGAS sensing of self-DNA. STIM1-mediated Ca^2+^ entry regulates both innate and adaptive immunity. However, studies indicate that STIM1 can regulate STING signaling independent of Ca^2+^. Thus, exploring the mechanism by which PIP, cGAS, and STIM1 are inked, and whether this pathway impacts cGAS-mediated self and foreign DNA are needed.

## Figures and Tables

**Figure 1 cells-11-02812-f001:**
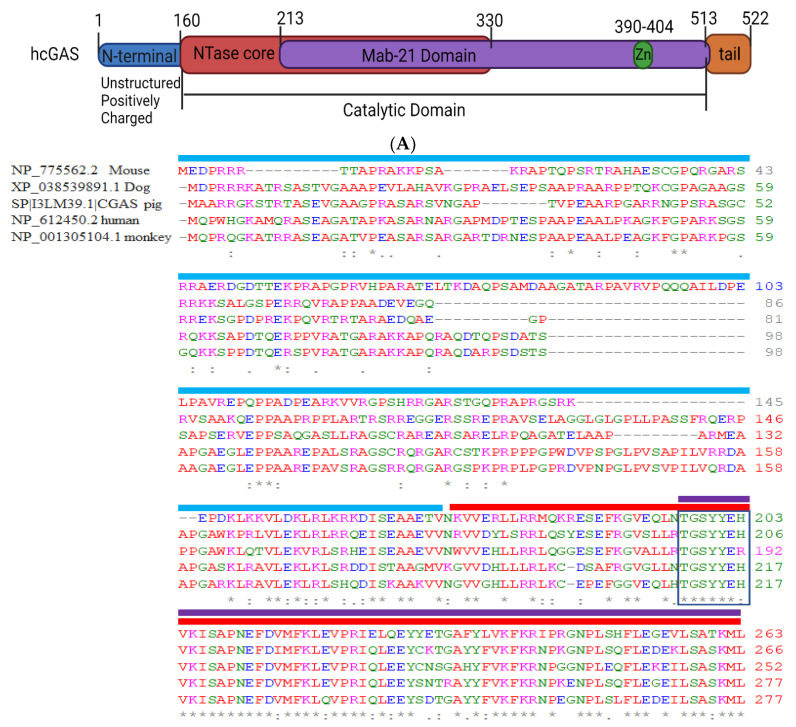
(**A**) Sequence structure of cGAS showing different domains. (**B**) Multiple sequence alignment of truncated amino acid sequences of (>NP_612450.2 cGAS-HUMAN, >NP_775562.2 cGAS-MOUSE, >NP_001305104.1 cGAS-MONKEY, >sp|I3LM39.1|CGAS-PIG, >XP_038539891.1 cGAS-DOG using Clustal OMEGA. cGAS protein sequence alignment from different mammalian species showed highly disorganized N terminal (~1–160 aa) whereas the C terminal of cGAS (161–370 aa) is more than 70% conserved among different mammalian species. The Blue box indicate the most conserved DNA binding region and the green box indicate the conserved Zinc Finger motif.

**Figure 2 cells-11-02812-f002:**
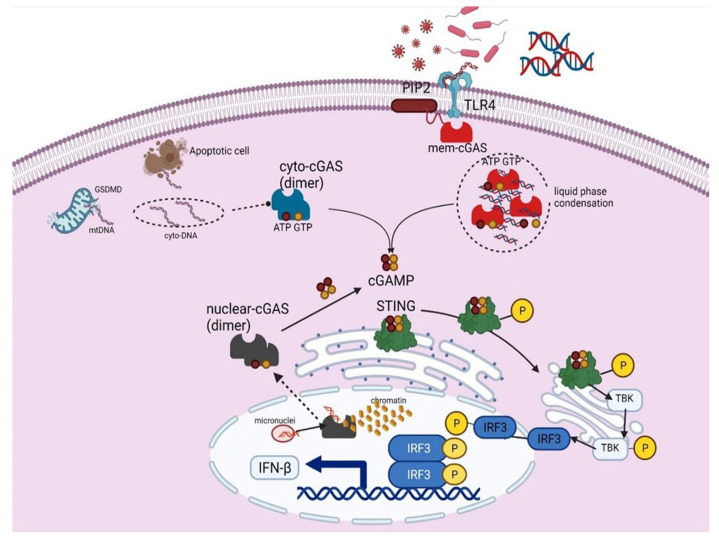
**cGAS localization and signaling.** cGAS is present in all the cellular compartments of the cell (membrane, cytosol, and nucleus). The positively charge N-terminal of cGAS binds with the negatively charged PIP2 present in the plasma membrane. Upon sensing dsDNA from pathogens or mt DNA excreting to the cytosol by forming a pore in mitochondria by GSDMD or ds DNA liberating from apoptotic cell or ds DNA from micronuclei or during chromatin repair during mitosis, cGAS dimerize which result in the generation of cGAMP. cGAMP binds to STING. STING autophosphorylates and translocates to the Golgi, where it phosphorylates TBK. *p*-TBK phosphorylate IRF3 which after phosphorylation enters the nucleus and results in the generation of IFN-β.

**Figure 3 cells-11-02812-f003:**
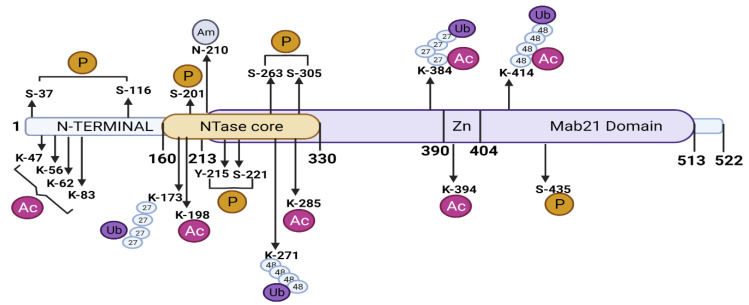
**Post-translational regulation of cGAS.** cGAS is regulated by various posttranslational modifications in response to dsDNA. The major post-translational modifications in the cells are monoubiquitination, polyubiquitination, phosphorylation, acetylation, and deamidation. This figure illustrates major post-translational modifications occurring on cGAS proteins.

**Figure 4 cells-11-02812-f004:**
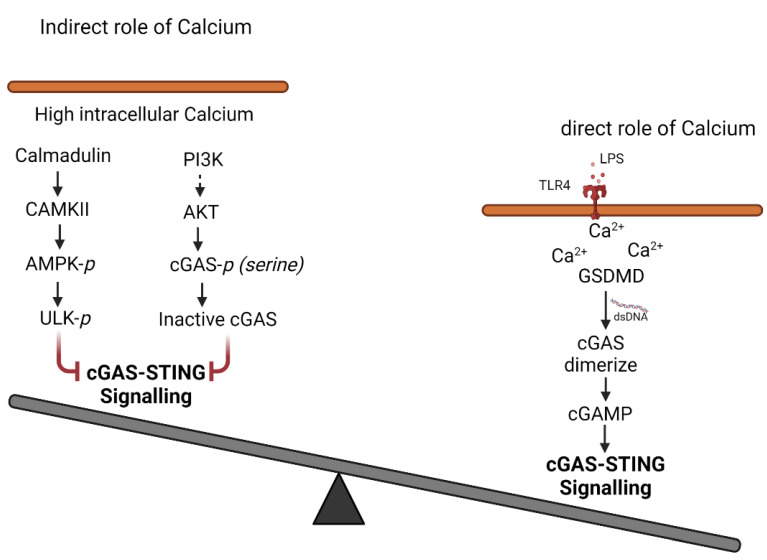
**Indirect role of calcium**. Calcium/calmodulin-dependent protein kinase II (CAMKII) may alter cGAS activation of STING. CAMKII induces phosphorylation of AMP-activated protein kinase (AMPK). Phosphorylation of AMPK induces STING signaling through repressing ULK1 by its phosphorylation. ULK1 phosphorylates and negatively regulates cGAS-STING signaling. Indirect activation of AKT via PI3K can also suppress cGAS activity through AKT phosphorylation of cGAS at serine residue (s305/s291). **Direct role of calcium**. LPS induces TLR4 signaling resulting in an increase in intracellular Ca^2+^. LPS also induce GSDMD cleavage, which in turn induces pore formation in the mitochondria releasing mitochondrial DNA (Mt DNA) in the cytosol. Mt DNA activates cGAS dimerization. This results in the generation of secondary messenger cGAMP and thus activation of cGAS-STING signaling.

## Data Availability

Not applicable.

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
