# Peer review of "Regulation of cGAS Activity and Downstream Signaling"

_cells, 2022, doi:10.3390/cells11182812_

Round 1
Reviewer 1 Report
Joshi et al. reviewed the available literature regulation of cGAS, which is predominantly expressed in various parts of the cells and majorly involved in the activation of innate immunity. More specifically, they examined the post-translational modification of cGAS upon binding dsDNA, its impact on STING signaling, and its role in developing immunity and cancers. Although the manuscript is nicely written, but there are still some shortcomings that require the authors’ attention.
1. Before jumping to the study's aims, please elaborate on the topic more with literature support, and tell your readers why you chose this topic.
2. Please provide concluding remarks for each section, simply saying what results of your analyses or highlights contribute to the overall conclusion.
3. Please add another section or paragraph for variability in cGAS sequence in different species; how about the overall function of different variants? It would be much of interest if you could highlight the functional differences based on sequence variation. How about splicing variants?
4. Please describe the mechanism of cGAS nuclear/cytosolic translocation for your readers. “Cytosolic cGAS may be beneficial for sensing genotoxic stimulation with the DNA that needs restraining in the nucleus or mitochondria” are you sure about mitochondria? Please provide a detailed description or evidence for mitochondrial stimulation.
5. Fig 3 some of the K residues of cGAS are shown to have Ubiquitination as well as acetylation. Could you explain more if both are happening at the same time? Just explain which modification occurs first. Let’s suppose AC happened first, then how UB will bind? In such conditions, how does cGAS behave? Interesting to see that only k27 and 48 ub occur, no other type of ub; please cross check!
6. Figure 4 is a total mess; the authors took the screenshot from the word file. Better rearrange this figure using some special software.
7. Cancer is a broad term, and authors should give a suitable title and cite the latest articles.
8. Please add a section on the role of cGAS in different diseases.
9. Add the clinical importance of cGAS and any clinical trials.
10. Conclusion and prospects should be written under separate headings.
11. Please add the critical questions to be solved.
Reviewer 2 Report
An interesting and comprehensive review that is largely in the final form. I would suggest the following changes:
Figure 1 is not very informative. This should include a structure of cGAS indicating the individual domains and their functions. At the moment the amino acid sequence is not very easy to follow.
Figure 3 legend should read "post-translational".
Figure 4 still has editing marks.
Round 2
Reviewer 1 Report
Most of my comments have been addressed adequately, and the quality of the manuscript has been improved. However, in the revised version, the authors forgot to provide authors' contributions and other statements.
Regarding the question, "Please add a section on the role of cGAS in different diseases" although authors have discussed "ALI, COVID-19...." I wanted to ask if the authors could provide additional literature support for different diseases (Other than they have provided already) related to cGAS signaling. The role of cGAS has been well studied in aging, liver, kidney, and neurological diseases. Dozens of review articles have already been published on these topics. For readers, convenience authors should highlight literature missing in previous studies or just summarize according to the theme of their study.
Please also cite the latest articles published.
Author Response
Dear reviewer,
Please see our response to “minor comments” from Reveiwer#1.
Most of my comments have been addressed adequately, and the quality of the manuscript has been improved. However, in the revised version, the authors forgot to provide the authors' contributions and other statements.
Thank you. We have added the author's contribution to the revised manuscript.
Regarding the question, "Please add a section on the role of cGAS in different diseases," although authors have discussed "ALI, COVID-19...." I wanted to ask if the authors could provide additional literature support for different diseases (Other than they have provided already) related to cGAS signaling. The role of cGAS has been well studied in aging, liver, kidney, and neurological diseases. Dozens of review articles have already been published on these topics. For readers, convenience authors should highlight literature missing in previous studies or just summarize according to the theme of their study. Please also cite the latest articles published.
In response, we have added a new section D4: Role of cGAS in other diseases (underlined). As the reviewer pointed out that several recent reviews have been published already on aging, liver, kidney, and neurological diseases. We have summarized these studies and cited the latest references.
Thank You
With Regards